# Improving Anytime Prediction with Parallel Cascaded Networks and a Temporal-Difference Loss

**Michael L. Iuzzolino**
Google Research, Brain Team
Department of Computer Science, University of Colorado

**Michael C. Mozer**
Google Research, Brain Team

**Samy Bengio**
Google Research, Brain Team[*]

## Abstract

Although deep feedforward neural networks share some characteristics with the primate visual system, a key distinction is their dynamics. Deep nets typically operate in *serial* stages wherein each layer completes its computation before processing begins in subsequent layers. In contrast, biological systems have *cascaded* dynamics: information propagates from neurons at all layers in parallel but transmission occurs gradually over time, leading to speed-accuracy trade offs even in feedforward architectures. We explore the consequences of biologically inspired parallel hardware by constructing cascaded ResNets in which each residual block has propagation delays but all blocks update in parallel in a stateful manner. Because information transmitted through skip connections avoids delays, the functional depth of the architecture increases over time, yielding anytime predictions that improve with internal-processing time. We introduce a temporal-difference training loss that achieves a strictly superior speed-accuracy profile over standard losses and enables the cascaded architecture to outperform state-of-the-art anytime-prediction methods. The cascaded architecture has intriguing properties, including: it classifies typical instances more rapidly than atypical instances; it is more robust to both persistent and transient noise than is a conventional ResNet; and its time-varying output trace provides a signal that can be exploited to improve information processing and inference.

Since the earliest investigations of artificial neural nets, their design has been informed by biological neural nets [37]. Perhaps the most compelling example is the convolutional net for machine vision, which has adopted properties of primate cortical neuroanatomy including a hierarchical layered organization, local receptive fields, and spatial equivariance [12]. In this article, we investigate computational consequences of two fundamental properties of biological information processing systems that have not been considered in the design of deep neural nets. First, *the brain consists of massively parallel, dedicated hardware with neurons throughout the cortex updating continuously and simultaneously.* Second, *information transmission between neurons introduces time delays* [1]. As a result, unrefined and possibly

---

[*]Currently at Apple.

35th Conference on Neural Information Processing Systems (NeurIPS 2021).

incomplete neural state in one region is transmitted to the next region even as the state evolves; and feedforward connectivity yields a speed-accuracy trade off in which the initial response to a static input occurs rapidly but can be inaccurate, with the output gradually improving over internal processing time. Following McClelland [36], we refer to such an architecture as *cascaded*.

Cascaded dynamics contrast sharply with the dynamics of standard feedforward nets, which operate in *serial* stages, each layer completing its computation before subsequent layers begin processing. Cascaded dynamics are also quite different than the dynamics of vision models with recurrent connections [e.g., 23, 25, 38, 47], which, given a static input, may iteratively update, but layer updates are still computed serially with each layer completing its computation and then feeding it immediately to the next layer (or back to itself). Fundamentally, our investigation asks: Supposing we take a step toward biological realism with massively parallel hardware and relatively slow inter-neuron communication, what are the computational benefits and consequences?[2]

We construct cascaded networks by introducing propagation delays in deep feedforward nets provided with a static input. We treat the net as massively parallel such that all units across all layers are updated simultaneously and iteratively. We focus on the ResNet architecture [14] and we introduce a propagation delay into each residual block (Figure 1a). Because the skip connection permits faster transmission of more primitive perceptual representations, the functional depth of the resulting architecture increases over internal-processing time, yielding a trade off between processing speed and complexity of processing. Consequently, the architecture offers a natural, integral mechanism for making predictions at any point in processing, known as *anytime prediction* [58]. Speed-accuracy trade offs are a fundamental characteristic of human information processing [22, 42] and human perception has been modeled with deep learning anytime prediction methods [29].

Although we focus on the ResNet, our approach can be incorporated into any model with skip connections (e.g., Highway Nets [48], DenseNet [19], U-Net [43], Transformers [52]). The contrast between a *serial*, one-layer-at-a-time model and a *cascaded*, parallel-update model is illustrated in Figures 1b and 1c, respectively. To step through the operation of the cascaded model, at time 1, only the first residual block has received meaningful input, and the model prediction is therefore based only on this block's computation. At time 2, *all* higher residual blocks have received input from block 1, and the output is therefore based on *all* blocks' computations, though blocks 2 and above have deficient input. At each subsequent time, all blocks are receiving meaningful input, but it is not until time $t$ that block $t$ has reached its asymptotic output because its input does not stabilize until $t - 1$. In essence, the cascaded model behaves like a WideResNet [56] on the first steps and then becomes a deep ResNet.

Our work makes the following key contributions.

- We demonstrate the superiority of the cascaded architecture to the serial (Figures 1b,c), indicating that parallelism can be exploited in a way that has not previously been studied.
- We propose and evaluate a novel training objective aimed at improving the predictions of anytime models. This *temporal-difference (TD) loss* [49] encourages the most accurate response as quickly as possible. TD training improves the performance of both cascaded and serial architectures. Although a rich literature exists aimed at reducing the number of computational steps required to obtain an accurate answer [2, 3, 4, 5, 11, 15, 16, 17, 18, 20, 24, 31, 31, 38, 40, 45, 51, 54, 57], all of this work uses a degenerate form of TD for training and our results suggest that these models can be improved using TD.
- The cascaded model trained with TD (CascadedTD) tends to respond most rapidly to prototypical exemplars, whereas training with the standard cross-entropy loss (CascadedCE) does not (Figure 2). We assess with three quantitative prototypicality measures, and we further show that CascadedTD rapidly converges on the correct semantic family, whereas CascadedCE does not. These facts indicate that CascadedTD organizes knowledge differently across layers than does CascadedCE.

---

[2]Like much other research in deep learning [25, 8], biology informs our work by providing novel forms of inductive bias. Our goal is to investigate computational consequences of these biases, not to model biological phenomena per se.

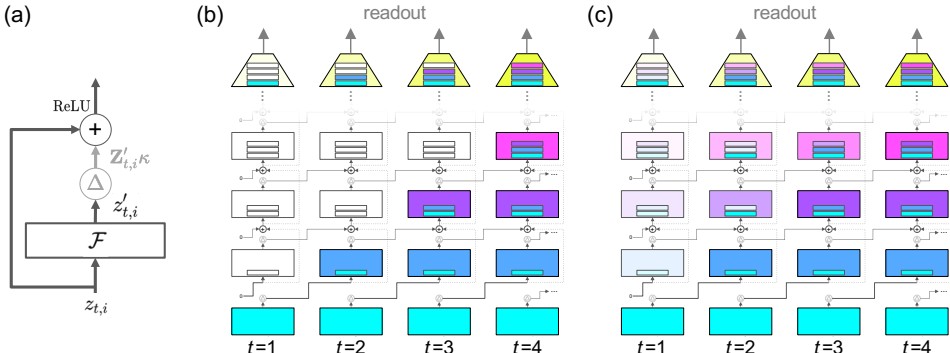

**Figure 1:** (a) ResNet building block, with additional delay component ($\Delta$, in grey) that convolves a temporal kernel with the block output. Details in text. (b) A standard *serial* ResNet is unrolled in time, with columns depicting time slices. Each rectangle is a ResNet block, which may consist of two or more convolutional layers. In the serial model, blocks are updated sequentially. Blocks which have not yet been activated are colored white and blocks which have been activated are shown in a hue unique to that block. The input is cyan. The narrow bars within each block signify the activation state of all blocks below that are contributing to the block's state (via skip connections). Read out from the model is via the yellow trapezoid at the top, which enables anytime prediction. The narrow bars inside the trapezoid indicate the block information available at each time for classification (via skip connections). (c) A *cascaded* ResNet is unrolled in time. In the cascaded model, all blocks update in parallel; however, at each step, they may rely on partial updates of lower blocks. As a result, multiple processing steps are required for a layer's activation to reach its asymptotic state. The color intensity (saturation) of a block indicates how close a block's activation state is to its asymptotic state.

- We show that CASCADEDTD obtains a strictly superior speed-accuracy profile compared to previously proposed anytime prediction models, which are all based on a serial architecture.
- We demonstrate other virtues of CASCADEDTD: it is more robust to input noise, and its time-varying output trace provides useful signals for *meta-cognitive* processes—separately trained nets that make judgments about the cascaded architecture's accuracy.

## Related Work

*Prior research on cascaded models.* From a psychological perspective, McClelland [36] characterizes human mental computation in terms of a hierarchy of leaky integrators that continually transmit partial information as it becomes available. We are aware of no work in deep learning on static image processing with cascaded models, but there exist two investigations focused on video sequence processing, where the model state from the previous frame is used to efficiently process the next. Fischer et al. [11] present a *streaming rollout* framework for recurrent nets and they very briefly explore the temporal dynamics of cascaded models, showing benefits to early predictions. They present a general taxonomy that includes our proposed feedforward cascaded model, but their focus is almost entirely on formal definitions and a framework that lays out the space of all well-formed roll out patterns (update orders). In contrast, our focus is almost entirely on training procedures that leverage the dynamics of cascaded models, on early read-out mechanisms, and on the computational consequences of these training and read-out mechanisms. Kugele et al. [28] focus on spiking neural net dynamics, on time-varying inputs, and on reductions in latency that are obtained as a sequence unfolds due to autocorrelations in the input sequence. Notably, Kugele et al. explore a variety of heuristic training losses and they settle on TD(1) as their preferred loss, but they do not explore the rest of the TD($\lambda$) family. Our work is complementary.

Carreira et al. [5] present a causal video understanding model that performs depth-parallel computation with the objective of improving video processing efficiency via maximizing throughput, minimizing latency, and reducing clock cycles.

*Recurrent nets for vision.* Recurrent nets have been used in vision [e.g., 20, 23, 27, 25, 32, 47, 46], which adds a dimension of internal processing time for every external input (see also

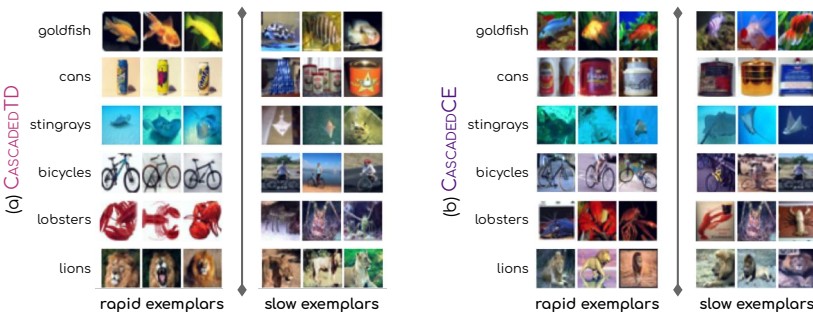

**Figure 2:** (a) CIFAR-100 instances categorized rapidly (left) and slowly (right) by a cascaded model trained using a TD loss. (b) same as (a) for a standard cross-entropy loss. The cascaded model with a TD loss stratifies instances by typicality, with rapid processing of prototypical views on a homogeneous background and slow processing of unusual and cluttered views.

[13]). However, these models perform serial layerwise updates and therefore fundamentally differ in their operation from cascaded models.

*Anytime prediction.* Anytime prediction models [2, 9, 15, 16, 17, 18, 20, 24, 30, 31, 34, 38, 40, 45, 51, 53, 54, 57] assume serial operation of layers, but allow for predictions to be made from intermediate layers of the architecture. In the simplest case, after $t$ steps, $t$ layers have been activated, and at each step, a prediction is made from the last activated layer [e.g., 16, 24]. Figure 1b illustrates a serial model that performs anytime prediction. Some of these models have intrinsic stopping criteria [e.g., 6]; others rely on selection of a stopping confidence threshold [e.g., 24, 51, 54].

*Temporal-difference learning.* TD learning has a rich history, mostly in the RL community for value function estimation. TD can be used for supervised learning as well. In fact, the two previous works in deep learning using cascaded models [5, 11] perform a boundary case of supervised TD, TD(1), which we show to have inferior performance. A variety of non-cascaded models, both recurrent [20, 31, 38, 57] and feedforward [2, 3, 4, 15, 16, 17, 18, 24, 31, 40, 45, 51, 54], aim to reduce the number of computational steps required to obtain an accurate output. All use TD(1) for training. No previous research has explored the general formulation of TD for improving anytime prediction.

## Deep Cascaded Networks

Many modern deep architectures—including ResNet [14], Highway Nets [48], DenseNet [19], U-Net [43]—incorporate skip connections that bypass strictly layered feedforward connectivity, analogous to the architecture of visual cortex [10]. Under the biological assumption that signals transmitted through a neural layer are delayed relative to signals that bypass the layer, we construct a cascaded model using ResNets by introducing a novel computational component that delays the transmission of signals from the output of each computational layer, denoted $\Delta$ in Figure 1a. Because these delays extend processing in time, the hidden states require a time index. The input to ResNet block $i$ at time $t$ is denoted $\boldsymbol{z}_{t,i}$. The block transforms this input via the residual transform, yielding $\boldsymbol{z}'_{t,i} = \mathcal{F}(\boldsymbol{z}_{t,i})$. We conceive of $\Delta$ as a tapped delay-line memory of the transform history, $\boldsymbol{Z}'_{t,i} = [\boldsymbol{z}'_{t,i} \; \boldsymbol{z}'_{t-1,i} \; \ldots \; \boldsymbol{z}'_{1,i}]$, which is convolved with a temporal kernel $\boldsymbol{\kappa}$ to produce the block output

$$\boldsymbol{z}_{t,i+1} = \text{ReLU}\left(\boldsymbol{z}_{t,i} + \boldsymbol{Z}'_{t,i}\boldsymbol{\kappa}\right). \tag{1}$$

The kernel $\boldsymbol{\kappa} = [1 \; 0 \; 0 \ldots 0]$ recovers the standard ResNet in which communication between layers is instantaneous. We consider two kernels to introduce time delays. With $\boldsymbol{\kappa} = [0 \; 1 \; 0 \; 0 \ldots 0]$, a discrete one-step delay is introduced (*OSD* for short). With $\boldsymbol{\kappa} = (1-\alpha)[1 \; \alpha \; \alpha^2 \; \alpha^3 \ldots]$, we obtain exponentially weighted smoothing (*EWS* for short), where larger $\alpha \in [0, 1)$ yield slower transmission times. Note that both of these special kernels have efficient implementations: the OSD kernel with a one-element queue and the EWS kernel with a finite (one-step) state vector and the incremental update,

$$\boldsymbol{Z}'_{t,i}\boldsymbol{\kappa} = \alpha \boldsymbol{Z}'_{t-1,i}\boldsymbol{\kappa} + (1-\alpha)\boldsymbol{z}'_{t,i}.$$

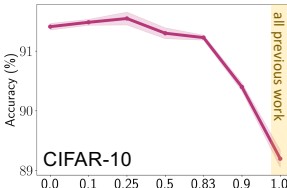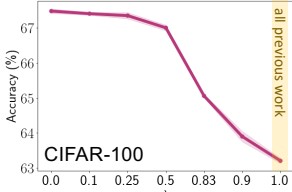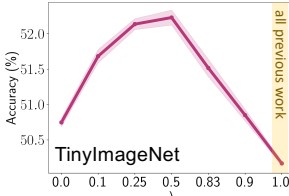

**Figure 3:** Effect of TD hyperparameter $\lambda$ on CASCADEDTD test accuracy for three data sets. $\lambda = 1$ corresponds to the training methodology of all past research on anytime prediction, which is inferior to any $\lambda < 1$ for all data sets. Shaded error bands—hard to see on most curves—indicate $\pm 1$ SEM, corrected to remove performance variance due to weight initialization and unrelated to $\lambda$ [35].

We use the OSD kernel for training all models. Modifications of batch norm were required to do time-step-conditional normalization (see Appendix A). All experiments use a ResNet-18, which has 8 residual blocks and hence 8 time delays. Note, we conducted experiments with larger ResNets and the additional compute did not affect qualitative properties. We also add a time delay to the output of the model's first convolutional layer. Consequently, with the OSD kernel, the cascaded model requires 9 updates for the output to reach asymptote. The cascaded and serial models with the same weights will necessarily produce identical asymptotic outputs.

To obtain a finer temporal granularity at evaluation, some simulations switch to the EWS kernel with $\alpha = 0.9$. Temporal dynamics are qualitatively similar for OSD and EWS, but EWS allows us to better distinguish individual examples in terms of their fine-grain timing. EWS with $\alpha = 0.9$ requires about 70 steps for the output to asymptote. We note that our findings are robust to the choice of $\alpha$, as long as $\alpha$ slows transmission.

**Training Cascaded Networks with TD($\lambda$)**

To allow for anytime prediction, we include an output head following each of the $T$ residual blocks in both the serial and cascaded models (see Figure 1b,c, respectively). The output heads may share weights or have separate weights. To encourage correct outputs sooner, we use temporal difference (TD) learning [49] over the output sequence. Readers may associate TD methods with reinforcement learning because TD methods have traditionally been used to predict future rewards. However, TD methods are fundamentally designed for supervised learning. We use TD to predict a future outcome—the correct classification of an image—from a sequence of successively more informative states—the information flowing through the ResNet at each internal time step.

TD($\lambda$) specifies a target output $y_t$ at each time $t \in \{1, ...T\}$ based on the model's actual output $\hat{y}_{t+i}$ at future times $t + i$ for $i > 0$, and the eventual outcome or true target, $y_{\text{true}}$:

$$y_t = (1 - \lambda) \left[ \sum_{i=1}^{T-t} \lambda^{i-1} \hat{y}_{t+i} \right] + \lambda^{T-t} y_{\text{true}}, \tag{2}$$

where $\lambda \in [0, 1]$ is a free parameter that essentially specifies the time horizon for prediction[3]. TD(1) predicts the eventual outcome at each step; TD(0) predicts the model's output at the next step (and the eventual outcome at the final step). Given target $y_t$ and actual output $\hat{y}_t$, we specify a cross-entropy loss, $\mathcal{L} = \sum_{t=0}^{T} H(y_t, \hat{y}_t)$, where $H(p, q)$ is the cross-entropy. Note that $y_t$ must be treated as a constant, not as a differentiable variable, via a `stop_gradient` (for TensorFlow or Jax) or `requires_grad=False` (for PyTorch). Although Equation 2 requires knowledge of all subsequent network states, the beauty of TD methods is that this loss can be computed incrementally (see Appendix A.2.1). The edge cases, TD(0) and TD(1), have trivial implementations. Past research has always used TD(1) for specifying intermediate targets, but we will show that TD(1) leads to local optima because the model is penalized for being unable to classify correctly at the earliest steps.

---

[3]We define $0^0 = 1$, as is generally agreed upon in the algebra community.

# Results

### TD($\lambda$) Training

We conducted a sweep over hyperparameter $\lambda$ to determine its effect on asymptotic accuracy of CASCADEDTD. Figure 3 shows results from five replications of CASCADEDTD on CIFAR-100, CIFAR-10, and TinyImageNet. The hyperparameter has a systematic effect, consistent with classic studies with linear models [50, Chapter 12]. The same effect is observed with high resolution images; see Appendix A.6, where we train a subset of ImageNet. Importantly, $\lambda = 1$, which is the implicit choice of *every* previous anytime-prediction model [2, 9, 15, 16, 17, 18, 20, 24, 30, 31, 34, 38, 40, 45, 51, 53, 54, 57], obtains the poorest performance for all data sets, significantly worse than $\lambda \approx .5$. The essential explanation is that larger $\lambda$ penalize the network for behavior it does not have the capability to achieve: obtaining the asymptotic prediction at the earliest time steps. To paraphrase the classic illustration of TD from Sutton [49], if the task is predicting the weather on December 31, no model can predict as accurately on December 1 as on December 30. Selecting $\lambda < 1$ shortens the prediction horizon; $\lambda = 0$ corresponds with requiring a prediction only of the weather on the next day.

In the rest of the article, we report results for CASCADEDTD with $\lambda = 0$, or *TD(0)*. Although TD(0) is not optimal for all data sets, it is strictly superior to TD(1) and has a trivial implementation because it does not require eligibility traces. Further, it avoids the need for a separate validation set to pick $\lambda$.

### Anytime Prediction and Speed-Accuracy Trade Offs

Given a static input, an anytime prediction model attempts to obtain the best classification possible as quickly as possible. Anytime prediction can be performed by both serial and cascaded models. Both yield predictions at each time slice, as depicted by the yellow trapezoids in Figures 1b,c, which denote model readout. Critical to anytime prediction is deciding when to terminate processing and initiate a response [24, 6, 51]. Following [24] and [51], we assume that processing terminates when the confidence (probability) for the most likely class rises above threshold $\theta$. For any $\theta$, one can measure the mean stopping time and the mean accuracy for all instances in a test set. By sweeping $\theta \in [0, 1]$ and plotting mean accuracy as a function of mean stopping time, one obtains a *speed-accuracy trade off* curve. Figure 4 shows curves for models we'll describe next. The horizontal axis indicates number of simulation time steps, which is linearly related to the matched number of operations (multiplies, additions, etc.) performed on simulated parallel hardware for all models. (See Appendix C for further details.)

To evaluate the cascaded model, we compare to a recent state-of-the-art method, the *Shallow-Deep Network (SDN)* [24]. The SDN has the serial architecture depicted in Figure 1b. One critical design decision was whether to have separate read-out heads at each step (MULTIHEAD) or a shared read-out head (SINGLEHEAD), i.e., whether weights are separate or shared. From the perspective of the cascaded model, which considers the vertical columns of Figure 1c to be copies of a network unrolled in time, the SINGLEHEAD approach is natural. The SDN, as a serial model, chose the MULTIHEAD approach. We tested all four logical combinations of {SERIALTD, CASCADEDTD} × {SINGLEHEAD, MULTIHEAD}. We use SERIALTD and CASCADEDTD as shorthand for the SINGLEHEAD variants, and append MULTIHEAD to the model name for that version. Additionally, we consider SERIALCE and CASCADEDCE, which are SINGLEHEAD variants trained with the standard cross entropy loss that penalizes only asymptotic accuracy and does not explicitly attempt to obtain a speeded response.

The key observations from Figure 4, which shows speed-accuracy trade offs for the six models on three data sets, are as follows. First, our canonical cascaded model, CASCADEDTD, obtains better anytime prediction than SERIALTD-MULTIHEAD (i.e., the architecture of SDN). CASCADEDTD also achieves higher asymptotic accuracy; its accuracy matches that of CASCADEDCE, a ResNet trained in the standard manner. Thus, cascaded models can exploit parallelism to obtain computational benefits in speeded perception without costs in

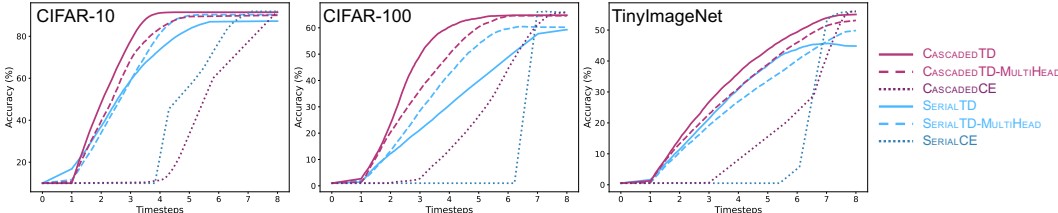

**Figure 4:** Speed accuracy trade off for three data sets and six models, obtained by varying a stopping threshold and measuring mean latency and mean accuracy. CASCADEDTD is our parallel anytime prediction model; SERIALTD-MULTIHEAD is the state-of-the-art method, SDN [24].

accuracy.[4] Second, while MULTIHEAD is superior to SINGLEHEAD for serial models, the reverse is true for cascaded models. This finding is consistent with the cascaded architecture's perspective on anytime prediction as unrolled iterative estimation, rather than, as cast in SDN, as distinct read out heads from different layers of the network. Third, models trained with TD outperform models trained with standard cross-entropy loss. Training for speeded responses reorganizes knowledge in the network so that earlier layers are more effective in classifying instances. We now turn to better understand what this reorganization entails.

**Organization of Knowledge in TD-Trained Cascaded Model**

Having examined the response profile of our models over an evaluation set, we now turn to analyzing the response to individual instances. Specifically, we ask about the time course of reaching a classification decision. We define the *selection latency* for an instance to be the minimum number of steps required to reach a confidence threshold on one class and maintain that level going forward, i.e., $\min\{t \mid [\exists j \mid \hat{y}_{t',j} > \theta \ \forall \ t' \geq t]\}$, where $\hat{y}$ is the model output, $j$ is an index over classes, and $\theta$ is the threshold. The selection latency does not specify whether or not the chosen class is correct. We picked a threshold $\theta = 0.83$ for CASCADEDTD such that only $\sim 10\%$ of the test examples failed to reach threshold; results that follow are robust to this selection.

What determines an instance's selection latency? Figure 2 presents instances that have the lowest and highest latency—labeled 'rapid' and 'slow', respectively. For CASCADEDTD (Figure 2a), notice the homogeneity of the rapid images: the objects are viewed from a canonical perspective and lie against a solid background with no clutter in the image. In contrast, the slow images are more varied, both in the object's instantiation in the image and the background complexity. Turning to CASCADEDCE (Figure 2b), instances do not appear to stratify by prototypicality. In the rest of this section, we formalize the notion of prototypicality with three measures and compute the correlation of each measure with selection latency for the cascaded model trained with a TD loss (CASCADEDTD) and with the standard cross-entropy loss (CASCADEDCE). Our three measures are as follows.

- *Centrality.* We compute the cosine distance of an instance's embedding (the penultimate layer activation) and the target-class weight vector. The larger this quantity, the better aligned the two vectors are. Because the weight vector will tend to point near the center of class instances, the cosine distance is a measure of instance centrality.
- *C-score.* Jiang et al. [21] describe an instance-based measure of statistical regularity called the C-score. The C-score is an empirical estimate of the probability that a network will generalize correctly to an instance if it is held out from the training set. It reflects statistical regularity in that an instance similar to many other instances in the training set should have a high C-score.
- *Human labeling consistency.* Peterson et al. [41] collected human labels on images. Most images are consistently labeled, but some are ambiguous. The negative entropy of the response distribution indicates inter-human labeling agreement. Presumably, consistently labeled instances are more prototypical.

---

[4]We used $\lambda = 1$ to train SERIALTD and SERIALTD-MULTIHEAD, as was done for *all* previously proposed serial models. Could SDN and other serial models be improved with $\lambda < 1$? In Appendix A.5.2, we show that the serial model with $\lambda = 0$ still does not perform as well as CASCADEDTD.

**Table 1:** Spearman rank correlations between three measures of instance prototypicality and selection latency for CIFAR-10. Prototypicality measures are formulated such that lower values correspond to higher prototypicality. Since the Spearman coefficient measures the degree to which prototypicality varies with selection latency, large coefficients indicate a correlation between fast responses and the prototypicality of an instances, whereas coefficients close to zero indicate no correlation. Here, we show that the prototypicality and selection latency of an instance for cascaded models are correlated, with CASCADEDTD yielding significantly higher correlations for two out of three measures as compared to CASCADEDCE.

| Measure | Spearman's $\rho$ | |
|---|---|---|
| | CASCADEDCE | CASCADEDTD |
| centrality | 0.140 | 0.352 |
| C-score | 0.326 | 0.489 |
| human consistency | 0.153 | 0.142 |

All three measures are available only for the CIFAR-10 training set. Consequently, we ran 10-fold cross validation on the training set, assessing the correlation based on the held out images in each fold. To obtain a granular selection latency, we use the EWS kernel.

Table 1 presents the correlation—Spearman's $\rho$—between the three prototypicality measures and negative selection latency for CASCADEDCE and CASCADEDTD. A positive coefficient indicates shorter latency for prototypical instances. The coefficient is reliably positive ($p < .001$) for each of the three typicality measures and both models. However, CASCADEDTD obtains reliably higher correlations than CASCADEDCE on two of the three measures ($p < .001$); they are not significantly different on the human consistency measure ($p = .29$). Thus, by these quantitative scores, the TD training procedure leads to better stratification of instances by typicality, in line with the qualitative results presented in Figure 2. Why does TD training distinguish instances based on prototypicality? Intuitively, a prototypical instance shares features with many other class instances. Because these features are frequent in the data set, the TD loss focuses on rapidly classifying instances with those features.

Beyond investigating the time course of fine-grain classification, we also examined coarse-grain classification. Forming twenty superclasses from the 100 fine-grain classes of CIFAR-100, as specified in [26], we examined the probability of correct coarse-grain classification conditional on incorrect fine-grain classification. Zamir et al. [57] refer to this probability as *taxonomic compliance*, which reflects information being transmitted about coarse category even when the specific class cannot be determined. As shown in Figure 5, taxonomic compliance rises faster for CASCADEDTD than for CASCADEDCE. Whereas chance compliance is .05, CASCADEDTD achieves a compliance probability of .35 after 2 steps. CASCADEDCE requires 8 steps to achieve the same performance. TD training pushes instances to the correct semantic neighborhood sooner, even when not to the correct class label. This result further supports the reorganization of knowledge for more robust decision making.

**Robustness to Input Noise**

In previous simulations, we have assumed the input was static while internal processing took place. Now we consider static inputs with time-varying noise. Figure 6 shows four types of lossy noise we consider on CIFAR-10 images. The noise types are: (1) *Focus*: a $16 \times 16$ foveated patch randomly placed within the image, where regions outside of the patch are Gaussian blurred; (2) *Perlin*: gradient noise randomly applied to 40% of image pixels; (3)

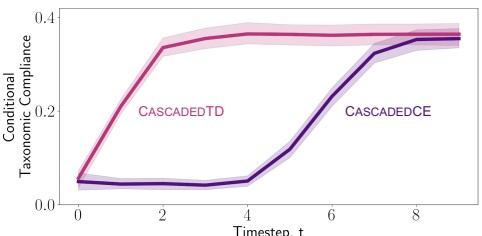

**Figure 5:** CASCADEDTD performs coarse-grain classification on CIFAR-100 before fine-train classification, as assessed by a measure of conditional taxonomic compliance [57]. Graphs are based on 5 runs of each model with different random initializations. Shaded error bands indicate $\pm 1$ SEM, corrected to remove performance variance due to initial seed [35].

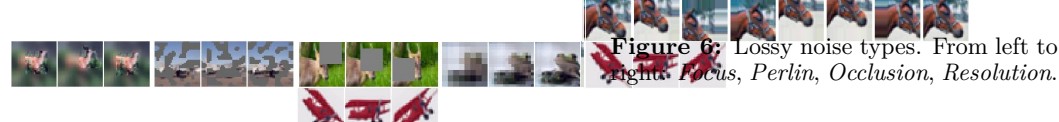

**Figure 6:** Lossy noise types. From left to right: *Focus, Perlin, Occlusion, Resolution.*

*Occlusion*: a $16 \times 16$ occluding patch randomly placed within the image; (4) *Resolution*: random downsampling by factors of $2\times$ or $4\times$ via average pooling followed by $k$-nearest upsampling to recover the original dimensionality of $32 \times 32$.

For each noise type, CASCADEDCE and CASCADEDTD models are trained with the corresponding image transformation type as a data augmentation. Because the external environment changes more rapidly than any snapshot of the environment can be processed, cascaded models will necessarily integrate signals from multiple snapshots. To determine whether signal integration is beneficial for noise suppression, we compared to a serial model that is allowed to fully process each snapshot, which for the architecture requires $9\times$ as many sequential updates as the cascaded model. We therefore refer to the model as SERIALCE$\times$9; it has the same weights as CASCADEDCE.

Two noise variants are applied to test images: *persistent*, in which a noise sample is drawn independently at each update step, and *transient*, which consists of three stages: first, the model reaches its asymptotic output on a noise-free input; second, the input is corrupted by noise samples for a variable number of steps; and third, the noise-free input is presented until the model returns to its previous asymptotic output. For persistent noise, we assess with asymptotic accuracy; for transient noise, we assess with a measure of *drop in integrated performance* over the course of the episode, which indicates how quickly the model recovers from noise perturbation (smaller is better). Simulation details provided in Appendix A.4.

Table 2 indicates that CASCADEDTD obtains a degree of robustness to persistent and transient noise not matched by the alternative models. Although SERIALCE$\times$9 performs a full inference pass on each noise perturbation, the stateful nature of CASCADEDTD allows it to smooth out noise via slow integration. Although CASCADEDCE shares the same architecture as CASCADEDTD, TD training is required to orchestrate the integration of content-specific perceptual information. This experiment indicates that laggy information processing in the cascaded model can be advantageous in a noisy environment. In Appendix A.4, we note that the benefit for CASCADEDTD applies only to lossy noise, not translations and rotations, as one would expect from the perspective of noise averaging.

**Meta-cognitive Inference**

In this section, we consider the hypothesis that temporally intermediate outputs from cascaded networks can provide additional signals to improve performance. We term this *metacognition*, by reference to human abilities to reason about our reasoning processes.

The temporal trace of output from CASCADEDTD is provided to a separate classifier, METACOG-OOD, which is *discriminatively* trained for out-of-distribution (OOD) detection. METACOG-OOD is a fully connected feedforward net with a 256-unit hidden layer and a sigmoidal output unit for binary prediction: 1 or 0 for in- or out-of-distribution instances, respectively. CIFAR-10's validation set serves as the in-distribution training examples, whereas the validation sets of TinyImageNet, LSUN, and SVHN serve as OOD training examples; see details in Appendix B.1. The CASCADEDTD output is represented in one of four ways as input to METACOG-OOD: (1) the confidence of the most probable class, known as the *max softmax prediction* (*MSP*),

**Table 2:** Experiments on persistent and transient input noise applied to CIFAR-10. Highlight indicates the best performance.

| | Persistent Noise | | | Transient Noise | | |
| | *Asymptotic Accuracy (%)* | | | *Drop in Integrated Performance* | | |
| **Noise** | SERIALCE$\times$9 | CASCADEDCE | CASCADEDTD | SERIALCE$\times$9 | CASCADEDCE | CASCADEDTD |
|---|---|---|---|---|---|---|
| Focus | $84.27 \pm 0.06$ | $83.75 \pm 0.10$ | $87.31 \pm 0.04$ | $0.62 \pm 0.04$ | $0.66 \pm 0.05$ | $0.00 \pm 0.01$ |
| Occlusion | $86.26 \pm 0.08$ | $82.73 \pm 0.09$ | $89.76 \pm 0.05$ | $7.70 \pm 0.55$ | $8.25 \pm 0.72$ | $0.93 \pm 0.15$ |
| Perlin | $85.18 \pm 0.03$ | $84.56 \pm 0.05$ | $87.67 \pm 0.08$ | $0.86 \pm 0.06$ | $0.87 \pm 0.07$ | $0.00 \pm 0.01$ |
| Resolution | $84.53 \pm 0.07$ | $85.40 \pm 0.07$ | $88.19 \pm 0.10$ | $0.81 \pm 0.06$ | $0.53 \pm 0.05$ | $0.18 \pm 0.02$ |

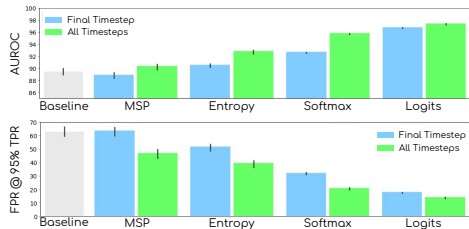

**Figure 7:** On CIFAR-10, the output of Cascaded­TD over time provides a reliable signal for improving OOD detection over using just its asymptotic output. Larger AUROC and smaller FPR are better. All output representations benefit from the temporal trajectory. Error bars reflect ±1 SEM corrected to remove comparison-unrelated variance [35]. Baseline is the final max-softmax prediction.

(2) entropy of the class posterior distribution, (3) the class posterior distribution, and (4) the logit representation of the posterior. We investigate whether feeding the output of all time steps to MetaCog-OOD leads to improved prediction relative to feeding only the final asymptotic output. Only the latter information is available in a standard feedforward net.

Following [33], we assess OOD performance with *AUROC*, the area under the ROC curve, and *FPR @ 95% TPR*, the false positive rate at 95% true positive rate. A baseline metric is computed directly from the final max softmax predictions of the Cascaded­TD model, whereas the other metrics are based on the MetaCog-OOD model output. Figure 7 indicates that the temporal output trajectory of Cascaded­TD provides a valuable signal for OOD detection. Our goal here was not to propose a method for OOD detection, but merely to demonstrate that in principle, there is information about the input that is conveyed by the cascaded model's dynamics but that is not available in a traditional classifier's output.

## Discussion

We investigated a neglected biologically-motivated architecture in which the bottleneck in neural information processing is transmission delays, not the number of neurons that can update in parallel. We proposed a temporal-difference (TD) loss that yields improved speed-accuracy trade offs. We showed that this model beats the state-of-the-art anytime prediction method, partly because of the TD loss and partly because of the model dynamics. The cascaded model has many distinctive properties, including: it classifies prototypical instances more rapidly than outliers; it performs coarse-to-fine semantic processing in which general semantic categories are rapidly inferred even when specific class labels are not; it is able to moderate time-varying input noise; and the temporal trace of the model's output provides an additional signal that can be exploited to improve information processing beyond that provided by the asymptotic model output. Of course, these interesting properties come at a computational cost when parallel hardware is simulated on existing compute infrastructure. We see three directions in which cascaded nets have particular potential.

- For neuroscientists using deep nets as a model of human vision, cascaded nets are a better approximation to the dynamics of the neural hardware. The properties we investigate—neurons operate in parallel, neurons are stateful, and neurons are slow to transmit information—seem likely to have a critical impact on the nature of cortical computing. As one simple illustration, cortical feedback processes are often posited to be critical for explaining differences in processing efficiency of visual stimuli [e.g., 23, 47]. We have shown that these difference might be partly explained by feedforward cascaded dynamics.

- For hardware researchers, cascaded networks are a possible direction for the future design of AI hardware. It is a direction quite unlike modern GPUs and TPUs, one that exploits massively parallel albeit slow and possibly noisy information processing. Our success in showing strong performance from cascaded models, as well as a training procedure to obtain quick and accurate responses, should encourage research in this direction.

- For AI research in anytime prediction, we've shown that existing models can be improved with a TD($\lambda$) loss; all past research has adopted $\lambda = 1$, which we show to be inferior to $\lambda < 1$. For researchers who care little about cascaded models per se, cascaded models offer an intriguing method to train serial feedforward models. One can take a serial feedforward model, turn it into a cascaded model for training with TD methods, and then run it in serial mode. We've shown that TD training can improve asymptotic model accuracy while still providing anytime predictions due to inductive biases it imposes on the organization of representations.

## Acknowledgments

The authors thank Tyler Scott, Anirudh Goyal, Pradeep Shenoy, and particularly our three anonymous reviewers for thoughtful comments and feedback on earlier drafts of this work.

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
