# A  Experiment Details

## A.1  CASCADEDCE and CASCADEDTD Experiment Details

For all cascaded and serial models, we used a ResNet-18 for CIFAR-10, CIFAR-100, and TinyImageNet datasets. We experimented with deeper nets, up to ResNet-50, but found no differences in model behavior. The models were trained using data parallelism over 8 GPUs (see §A.7 for infrastructure details), with the model on each GPU using a batch size of 128. SGD with Nesterov momentum, an initial learning rate of 0.1, weight decay of 0.005, and momentum of 0.9 was used to optimize a softmax cross-entropy loss for SERIALCE/CASCADEDCE and a temporal difference cross-entropy loss for SERIALTD/CASCADEDTD. All models were trained for 120 epochs and the learning rate was decayed with a multiplicative factor of 0.2 every 30 epochs.

The training datasets were split (class-balanced) as 90-10 train-validation, where the validation splits were held out for downstream tasks, such as when training the meta-cognitive models (see Appendix B). For CASCADEDTD, the batch normalization layer must be augmented such that running means and variances are tracked independently for each timestep. At run-time, if the maximum number of timesteps used during training is exceeded, as occurs when using the EWS kernel, the final timestep statistics of the batch normalization layers are used for all subsequent timesteps. Furthermore, we observed that the offset parameter of the affine transformation of the batch normalization on identity mappings blows up to NaN values during training; consequently, we do not use batch normalization on the identity mapping in the cascaded nor serial models.

## A.2  Temporal Difference Loss

### A.2.1  Incremental TD Formulation

TD($\lambda$) amounts to training at each time step $t$ with a target, $y_t^\lambda$, that is an exponentially decaying trace of future outputs, anchored beyond some asymptotic time $T$ to the true target, $y$. Denoting the network output at step $t \in \{1, \ldots, T\}$ as $\hat{y}_t$, the trace is:

$$y_t^\lambda = (1 - \lambda) \sum_{n=1}^{\infty} \lambda^{n-1} \bar{y}_{t+n}$$

$$\text{with} \ \ \bar{y}_{t+n} = \begin{cases} \hat{y}_{t+n} & \text{if } t + n \leq T \\ y & \text{otherwise} \end{cases}$$

$$= (1 - \lambda) \sum_{n=1}^{T-t} \lambda^{n-1} \hat{y}_{t+n} + \lambda^{T-t} y.$$

We train with cross entropy loss over all time steps. For a single example, the loss is

$$\mathcal{L} = -\sum_{t,i} y_{ti}^\lambda \ln \bar{y}_{ti}.$$

The derivative of this loss with respect to the network parameters $w$ can be expressed in terms of the derivative with respect to the logits:

$$\nabla_w \mathcal{L} = -\sum_{t,i} (y_{ti}^\lambda - \bar{y}_{ti}) \nabla_w z_{ti},$$

where $z_{ti}$ is the logit of class $i$ at step $t$. The temporal difference method provides a means of computing this gradient incrementally, such that at each step $t$, an update can be computed based on only the difference of model outputs at $t$ and $t + 1$:

$$\nabla_w^{\text{TD}} \mathcal{L} = -\sum_{t,i} (\bar{y}_{t+1,i} - \bar{y}_{ti}) e_{ti},$$

where $e_{ti}$ is an *eligibility trace*, defined as:

$$e_{ti} = \begin{cases} \mathbf{0} & \text{if } t = 0 \\ \lambda e_{t-1,i} + \nabla_w z_{ti} & \text{if } t \geq 1 \end{cases}$$

**Table A.1:** Asymptotic accuracy for CASCADEDTD models for various $\lambda$, as well as CASCADEDCE. Green font indicates best performance across TD($\lambda$) and CASCADEDCE models for a given dataset. Highlight indicates best performing TD($\lambda$) across $\lambda$'s for a given dataset.

| Dataset | Cascaded Model Variant | | | | | |
|---|---|---|---|---|---|---|
| | TD(0) | TD(0.25) | TD(0.5) | TD(0.83) | TD(1) | CE |
| CIFAR-10 | $91.22 \pm 0.18$ | $91.65 \pm 0.08$ | $91.45 \pm 0.16$ | $90.98 \pm 0.21$ | $88.75 \pm 0.42$ | $91.91 \pm 0.08$ |
| CIFAR-100 | $67.48 \pm 0.14$ | $67.35 \pm 0.20$ | $67.00 \pm 0.18$ | $65.06 \pm 0.11$ | $63.20 \pm 0.14$ | $65.56 \pm 0.06$ |
| TinyImageNet | $50.74 \pm 0.11$ | $52.03 \pm 0.07$ | $52.25 \pm 0.07$ | $51.39 \pm 0.13$ | $49.86 \pm 0.15$ | $52.46 \pm 0.06$ |

The incremental formulation of TD via $\nabla_w^{\text{TD}}\mathcal{L}$ is valuable when gradients and/or weight updates must be computed on line rather than presenting an entire sequence before computing the loss, e.g., in the situation where the network runs for many steps and truncated BPTT is required. In our experiments, we use the summed gradient, $\nabla_w\mathcal{L}$, computed by PyTorch from the full $T$ step sequence and our exponentially weighted target, $y_t^\lambda$.

### A.2.2 TD($\lambda$) Sweep

Table A.1 shows the tabulated results for asymptotic accuracy of CASCADEDTD swept over $\lambda$ values on CIFAR-10, CIFAR-100, and TinyImageNet, as well as CASCADEDCE. Note, 5 trials per $\lambda$ were trained for each dataset.

### A.3 Data Augmentation

When training all models on CIFAR-10 and CIFAR-100, for each batch the $32 \times 32$ images are padded with 4 pixels to each border (via reflection padding), resulting in a $40 \times 40$ image. A random $32 \times 32$ crop is taken, the image is randomly flipped horizontally, and standard normalized using the training set statistics is applied. Finally, a random $8 \times 8$ block cut is taken such that the cropped pixels are set to 0. Images at run-time are only standard normalized using training set statistics - no other augmentation is applied with the exception of the persistent and noise robustness experiments. The same process is followed for TinyImageNet with the following exceptions: (1) the $64 \times 64$ images are padded to $86 \times 86$ with reflection padding, random cropped back to $64 \times 64$, randomly flipped horizontally, then standard normalized, and (2) no $8 \times 8$ block cutting is applied.

### A.4 Noise Experiments

The four noise perturbations studied in the main article are lossy. Here we consider two additional noise sources that are roughly information preserving: *Translation*: random shifts $\pm 8$ pixels in $(x, y)$ on a reflection-padded image; *Rotation*: random rotations $\pm 60°$ on a reflection padded image.

SERIALCE×9 is best on information preserving transformations such as *Translation* and *Rotation* because it is performing a full inference pass on the input whereas the cascaded models are performing a single update step.

While both CASCADEDTD and CASCADEDCE smooth responses over frames, CASCADEDTD performs better, indicating that beyond smoothing, TD training orchestrates the integration of image-specific perceptual information. This integration matters more for lossy transformations, where information integration is essential.

The training details are the same as previous simulations, except that we discard the $8 \times 8$ block data augmentation in order to avoid biasing the models toward the *Occlusion* noise transformation. We evaluate the cascaded models with the OSD kernel to allow for a comparison of cascaded models with SERIALCE×9.

In the persistent-noise experiment, five trials are run per image in the test set. In the transient-noise experiment, we present the noise-free input for 10 time steps (sufficient for the cascaded models to reach asymptote), apply one of the six noise transforms for $N$ time steps, and then present the noise-free input for another 10 steps, allowing the model to return to its asymptotic state. We run five trials per condition for each $N \in \{1, 2, 3, 4, 5, 6\}$ and each image in the test set. Performance is evaluated as the drop-in-integrated-performance,

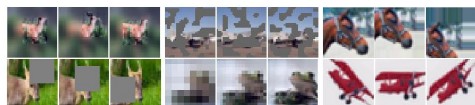

**Figure A.1:** Noise types. From left to right, top to bottom: *Focus, Perlin, Translation, Occlusion, Resolution, Rotation.*

**Table A.2:** Persistent-noise experiment. Highlight indicates best asymptotic performance for a given noise type.

| Persistent | **Asymptotic Model Performance** | | |
|---|---|---|---|
| **Noise** | SERIALCEx9 | CASCADEDCE | CASCADEDTD |
| Focus | $84.27 \pm 0.06$ | $83.75 \pm 0.10$ | $87.31 \pm 0.04$ |
| Occlusion | $86.26 \pm 0.08$ | $82.73 \pm 0.09$ | $89.76 \pm 0.05$ |
| Perlin | $85.18 \pm 0.03$ | $84.56 \pm 0.05$ | $87.67 \pm 0.08$ |
| Resolution | $84.53 \pm 0.07$ | $85.40 \pm 0.07$ | $88.19 \pm 0.10$ |
| Rotation | $89.11 \pm 0.04$ | $73.79 \pm 0.10$ | $87.51 \pm 0.03$ |
| Translation | $87.55 \pm 0.12$ | $76.72 \pm 0.09$ | $83.42 \pm 0.14$ |

**Table A.3:** Transient-noise experiment. Highlight indicates lowest DIP for a given noise type.

| Transient | **Drop in Integrated Performance (DIP)** | | |
|---|---|---|---|
| **Noise** | SERIALCEx9 | CASCADEDCE | CASCADEDTD |
| Focus | $0.62 \pm 0.04$ | $0.66 \pm 0.05$ | $0.00 \pm 0.01$ |
| Occlusion | $7.70 \pm 0.55$ | $8.25 \pm 0.72$ | $0.93 \pm 0.15$ |
| Perlin | $0.86 \pm 0.06$ | $0.87 \pm 0.07$ | $0.00 \pm 0.01$ |
| Resolution | $0.81 \pm 0.06$ | $0.53 \pm 0.05$ | $0.18 \pm 0.02$ |
| Rotation | $0.24 \pm 0.02$ | $4.12 \pm 0.29$ | $0.00 \pm 0.01$ |
| Translation | $0.72 \pm 0.05$ | $4.17 \pm 0.37$ | $1.53 \pm 0.14$ |

$\text{DIP} = \hat{y}_T - \mathbb{E}_{t \in \{B,\dots,T\}}[\hat{y}_t]$, where $T$ is the total time steps in the simulation, $B$ is the onset time of the noise transformations, and $\hat{y}_t$ is the model's target-class confidence at time step $t$. DIP indicates how quickly a model can recover from noise perturbations.

## A.5 Additional Temporal Dynamics Results

### A.5.1 Deadline-based stopping criterion

In the main paper, we show speed-accuracy trade offs for models based on a stopping criterion that terminates processing when a *confidence threshold* is reached for one output class. In Figure A.2, we examine an alternative stopping criterion that is based on a *temporal deadline*, i.e., after a certain number of update iterations. When the speed-accuracy curves for the two stopping criteria are directly compared, the confidence-threshold procedure is superior for all models. For this reason, we report the confidence-threshold procedure in the main paper. However, the confidence-threshold procedure does not allow us to readily compute error bars across model replications because the mean stopping time is slightly different for each

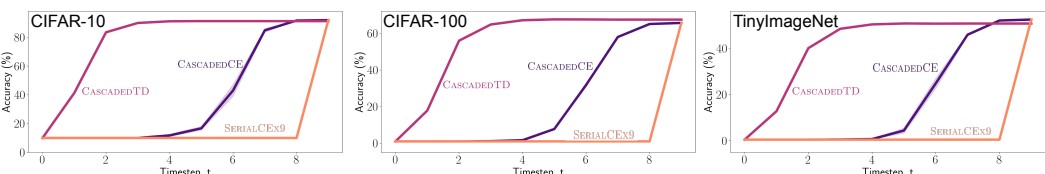

**Figure A.2:** Speed accuracy trade off for CIFAR-10, CIFAR-100, and TinyImageNet. Here we show the dynamics for a temporal-deadline stopping criterion, whereas Figures 4 and A.3 show speed accuracy trade offs obtained by a confidence threshold-based criterion. Accuracy assuming a particular stopping time is computed for each deadline. Note, SERIALCE×9 produces an output only after all updates. Confidence intervals across model replications are shown by the shaded regions, which are difficult to see because the uncertainty is small.

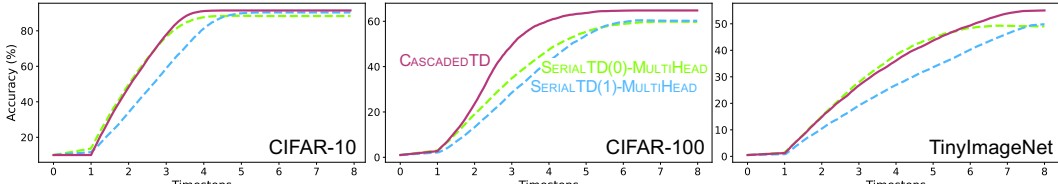

**Figure A.3:** Comparison of CASCADEDTD, SERIALTD(1)-MULTIHEAD (Shallow-Deep Nets), and SERIALTD(0)-MULTIHEAD. Shallow-Deep Nets can be improved using TD(0), but the speed-accuracy trade off is still significantly worse on CIFAR-100, and asymptotic accuracy is lower than that of CASCADEDTD.

replication. In Figure A.2, we show confidence intervals at the various stopping times using shaded regions. (The regions are very small and are difficult to see.) The main reason for presenting these curves is to convince readers of the reliability of the speed-accuracy curves.

### A.5.2 Serial models trained with TD

In the main paper, we compare cascaded models to Shallow-Deep Networks, a serial model with multi-headed outputs trained with TD(1). Just as training with $\lambda < 1$ improves performance of CASCADEDTD, one might hope to observe a similar benefit for serial models such as Shallow-Deep Networks. Figure A.3 shows that indeed training with TD(0) is superior to training with TD(1) for SDNs, labeled in the graph as SERIALTD-MULTIHEAD. The serial model's performance improves significantly, nearly to the level of CASCADEDTD, for two data sets, but for the third, CASCADEDTD still has a considerable advantage over the serial model, whether trained with $\lambda = 0$ or $\lambda = 1$.

### A.5.3 Qualitative performance of TD trained models on CIFAR-10

Figure A.4 shows CIFAR-10 instances with low and high *selection latency* for both CASCADEDTD and CASCADEDCE models. As with CIFAR-100, the qualitative differences between low and high selection latency for CASCADEDTD are stark, with low selection latency instances being more representative of prototypical instances of the given class (e.g., boats on blue water; horses in fields), whereas high selection latency instances are less typical (e.g., boats on green grass; horses in snow). In contrast, the strong delineation between low and high selection latency groups is not observed for CASCADEDCE, supporting the claim that TD training allows the model to more rapidly respond to prototypical exemplars.

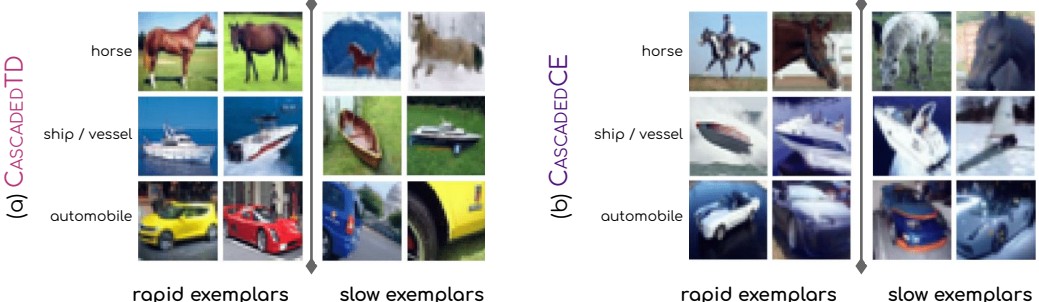

**Figure A.4:** (a) CIFAR-10 instances categorized raplidy (left) and slowly (right) by a cascaded model trained using a TD loss. (b) same as (a) for a standard corss-entropy loss. As observed for CIFAR-100, the cascaded model trained with a TD loss on CIFAR-10 stratifies instances by typicality, with rapid processing of prototypical views on a homogeneous background and slow processing of unusual and cluttered views.

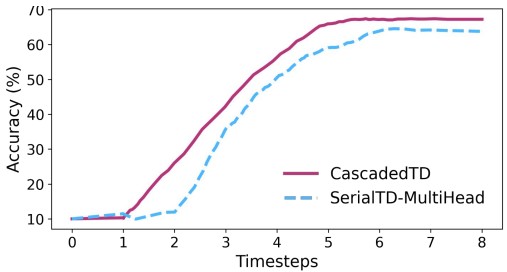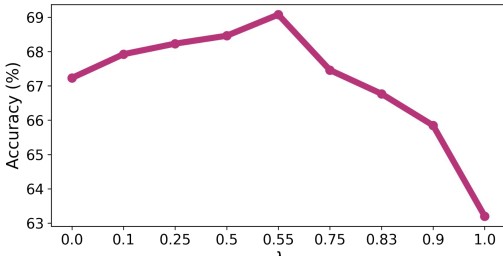

**Figure A.5:** ImageNet2012 Results. Left: Speed accuracy trade off for terrier breed subset of ImageNet2012, obtained by varying a stopping threshold and measuring mean latency and mean accuracy. CASCADEDTD is our parallel anytime prediction model; SERIALTD-MULTIHEAD is the state-of-the-art method, SDN [24]. Right: Effect of TD hyperparameter $\lambda$ on CASCADEDTD test accuracy. $\lambda = 1$ corresponds to the training methodology of all past research on anytime prediction, which is inferior to any $\lambda < 1$.

## A.6    Generalization to High Resolution Images

To assess how well our methods generalize to high resolution images, we trained a model on $224 \times 224$ resolution images of 10 breeds of terriers (dogs) from ImageNet2012 [44]. These breeds are visually similar to one another and cannot be discriminated perfectly based on any simple feature such as color. We observed the same qualitative behavior from models as we observed for our models trained on smaller resolution images (see Figure A.5 (left)). We obtain an asymptotic test accuracy of 67.2% for CASCADEDTD(0) versus 63.2% for SERIALTD(1)-MULTIHEAD (i.e., Shallow-Deep Nets), and a 13% speed up for CASCADEDTD(0) to reach a threshold of 50% accuracy; see Figure A.5 (left).

We conducted a sweep over hyperparameter $\lambda$ to determine its effect on asymptotic accuracy of CASCADEDTD. Figure A.5 (right) shows results for our terrier subset of ImageNet2012. Our results here are consistent with those from our smaller resolution dataset experiments from the main paper, where the $\lambda$ hyperparameter has a systematic effect on performance such that $\lambda < 1$ yields significantly improved performance. For example, TD(0.55) achieves a performance of 69.1%, which is a 9.3% improvement over TD(1). This supports our claim that our method generalizes to higher resolution datasets.

For training details, each class consisted of 1300 examples, which were split 90/10 into train and test examples. All simulation details were the same as our previous work, except batch size had to be reduced from 128 to 32 for the larger images, and, the first convolution layer of the ResNet was changed from 3x3 to 7x7 receptive fields.

## A.7    Computing Infrastructure

We used 8x NVIDIA Tesla V100's on Google Cloud Platform (GCP) for training all CAS-CADEDCE and CASCADEDTD models; a single V100 was used for all evaluations, and to train METACOG-OOD and METACOG-RESP models. All models were implemented in PyTorch v1.5.0, using Python 3.7.7 operating on Ubuntu 18.04.

**Table A.4:** Average runtime for training CASCADEDCE and CASCADEDTD over CIFAR-10, CIFAR-100, and TinyImageNet.

| Dataset | Model | Average Runtime (hours) |
|---------|-------|--------------------------|
| CIFAR-10 | CASCADEDCE | $1.48 \pm 0.002$ |
| | CASCADEDTD | $1.81 \pm 0.001$ |
| CIFAR-100 | CASCADEDCE | $1.48 \pm 0.001$ |
| | CASCADEDTD | $1.83 \pm 0.001$ |
| TinyImageNet | CASCADEDCE | $1.45 \pm 0.035$ |
| | CASCADEDTD | $1.97 \pm 0.020$ |

## A.8 Average Runtime and Reproducibility

Table A.4 shows average run times (in hours) for CASCADEDCE and CASCADEDTD. Variability in run time, expressed as $\pm 1$ SEM, is also shown. Reproducibility was ensured in the data pipeline and model training by seeding Random, Numpy, and PyTorch packages, as well as flagging deterministic cudnn via PyTorch API. When sweeping over $\lambda$ for a given model and dataset, 5 replications were trained to obtain reliability estimates; a fixed set of 5 seeds was for all models to ensure matched initial conditions across models. The average runtime for training the meta-cognitive models on a single V100 GPU requires less than 3 minutes. When training multiple trials for a given meta-cognitive model, all models are initialized with the same weights, and 42 was used to seed all packages as detailed above.

## B   Meta-cognitive Experiment Details

For all meta-cognitive experiments, training data is generated from the EWS kernel applied to CASCADEDTD(0).

## B.1 OOD Detection Dataset Details

CIFAR-10 serves as the in-distribution dataset, which contains 5,000 validation and 10,000 test set instances. The 5,000 validation instances, which we use as the in-distribution training set for OOD, were derived from a 90-10 train-validation split of the original 50,000 training instances used for training the CASCADEDTD model. The OOD datasets are as follows:

**TinyImagenet** The Tiny ImageNet (TinyImageNet) is a 200-class subset of ImageNet [7] and it contains 10,000 validation and 10,000 test instances. Following the methods of [33] we introduce two variations: 1) *resize*; the image is downsampled to $32 \times 32$, and 2) *crop*; a random $32 \times 32$ crop is taken from the image.

**LSUN** The Large-scale Scene UNderstanding (LSUN) [55] consists of 10 scenes categories, such as classroom, restaurant, bedroom, etc. It contains 10,000 validation and 10,000 test instances, and similar to TinyImageNet, we use the *resize* and *crop* variations.

**SVHN** The Street View House Numbers (SVHN) [39] dataset is obtained from house numbers in Google Street View images. It consists of 73,257 validation and 26,032 test set images.

## B.2 OOD Detection Training Details

The METACOG-OOD model is trained for 300 epochs with batch sizes of 256. We used Adam with an initial learning rate of 0.001 and weight decay of 0.0005 to optimize a binary cross entropy loss. Dropout with keep probability 0.5 was used for regularization. Numerical values corresponding to Figure 7 are tabulated in Table B.5 with reported SEM corrected to remove random variance [35].

The OOD examples from TinyImageNet and LSUN have crop and resize variations [33] to make them match CIFAR10 images in dimensions. METACOG-OOD is trained per (in-, out-of-distribution) dataset pairing—e.g., (CIFAR-10, SVHN)—and input representation type (discussed below). The respective test set is used for evaluation.

## B.3 Response Initiation

We explored a third stopping criterion, in addition to the confidence-threshold and temporal-deadline criteria. The third criterion was based on a meta-cognitive model that observes the output *sequence* from cascaded-model updates and uses this sequence to determine when to stop. To handle sequences, this METACOG-RESP model was an RNN, specifically a GRU, trained with a logistic output unit that produced a binary stop/don't-stop decision. In contrast to the confidence-threshold criterion, which is based solely on the network output at step $t$, METACOG-RESP in principle uses steps $0 - t$ to make its decision. It produces a continuous output in [0,1], and by stopping when the output rises above a threshold, we can

**Table B.5:** CIFAR-10 (in-distribution) vs. Aggregate OOD dataset quantitative measures corresponding to Figure 7. Each representation may include all time step outputs, $t_{\text{all}}$, or only the final output, $t_{\text{final}}$.

| OOD Representation | AUROC | FPR @ 95% TPR |
|---|---|---|
| CASCADEDTD [MSP] | $89.5 \pm 0.5$ | $63.0 \pm 3.5$ |
| METACOG-OOD $t_{\text{final}}$ [MSP] | $88.8 \pm 0.5$ | $63.0 \pm 3.1$ |
| METACOG-OOD $t_{\text{all}}$ [MSP] | $90.2 \pm 0.5$ | $46.4 \pm 3.1$ |
| METACOG-OOD $t_{\text{final}}$ [Entropy] | $90.5 \pm 0.3$ | $51.2 \pm 2.5$ |
| METACOG-OOD $t_{\text{all}}$ [Entropy] | $92.7 \pm 0.3$ | $38.9 \pm 2.5$ |
| METACOG-OOD $t_{\text{final}}$ [Softmax] | $92.6 \pm 0.1$ | $31.7 \pm 0.7$ |
| METACOG-OOD $t_{\text{all}}$ [Softmax] | $95.7 \pm 0.1$ | $20.5 \pm 0.7$ |
| METACOG-OOD $t_{\text{final}}$ [Logits] | $96.7 \pm 0.1$ | $17.5 \pm 0.4$ |
| METACOG-OOD $t_{\text{all}}$ [Logits] | $97.3 \pm 0.1$ | $13.7 \pm 0.4$ |

map out a speed-accuracy trajectory analogous to that obtained with the confidence-threshold criterion.

METACOG-RESP is trained for 300 epochs with a batch size of 256. We used Adam with an initial learning rate of 0.0001 and weight decay of 0.0001 to optimize a binary cross entropy loss. The supervised target is 1.0 at step $t$ if the output with highest probability at $t$ remains unchanged for all subsequent steps, or 0.0 otherwise. Essentially, the model is trained to predict when additional compute will change its decision. To obtain a finer granularity on time steps, we trained and evaluated METACOG-RESP with the EWS kernel.

We trained METACOG-RESP to predict when to stop for both CASCADEDCE and CASCADEDTD. METACOG-RESP is trained on the 4,500 instances of the CIFAR-10 validation set that have been processed by the cascaded model, yielding a training set of dimension $4,500 \times 70 \times 10$, where there are 70 timesteps and 10 logit values. We generate our evaluation set from the same method above using the CASCADEDCE model on the 10,000 instance test set.

Figure B.1 shows the response initiation results comparing CASCADEDTD (left panel) and CASCADEDCE (right panel) with stopping criterion using METACOG-RESP versus a temporal deadline. The METACOG-RESP criterion yields significant improvements to response initiation for both models. This finding lends support to the notion that there is a signal in the model output over time as information trickles through the cascaded layers. Essentially, METACOG-RESP can interpret the temporal evolution of cascaded model outputs to improve its speed-accuracy trade off.

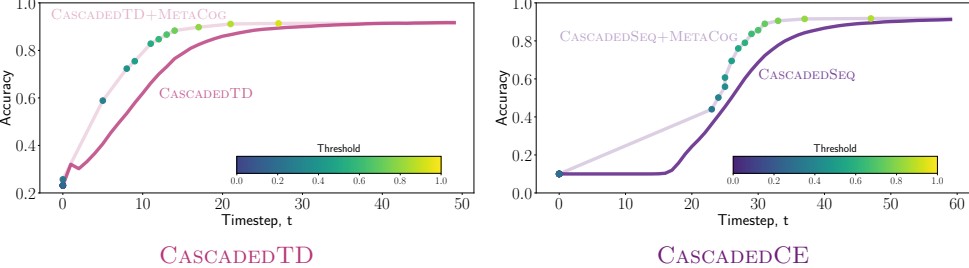

**Figure B.1:** Response initiation results comparing two stopping criteria for CASCADEDTD (left panel) and CASCADEDCE (right panel). The solid line represents a temporal-deadline stopping criterion. The fainter dotted line uses METACOG-RESP to determine when to stop based on an output threshold.

## C  Correspondence Between Model Time Steps and Run Time on Parallel Hardware

The premise of our work is that we have massively parallel hardware with delays on inter-component communication. This architecture allows all ResNet blocks to be updated simultaneously. A block update involves a sequence of matrix multiplications, vector

additions, and vector thresholding operations. Because all model variants perform block updates, we needn't break down the block update into its primitive operations; instead, we consider the basic cycle to be a block update. Due to the assumed communication delay in our hardware, the updated output from a block is not available to other blocks until the next cycle. Both the Serial and Cascaded models can be run on this parallel hardware. The Serial model does not take full advantage of the parallel hardware because it updates only block $k$ at cycle $k$. The Serial model performs anytime read out at cycle $k$ by summing the outputs of blocks 1 through $k-1$ and passing the sum through the classifier head. The Cascaded model takes full advantage of the parallel hardware by updating all blocks at each cycle $k$ using their output from cycle $k-1$. Just as the Serial model, the Cascaded model can perform any-time read out using the one-cycle lagged block outputs. Thus, comparing the Serial and Cascaded models in units of block updates—as we have done throughout the article—provides a runtime comparison on the assumed parallel hardware. Note that Fischer et al. [11] similarly compare models with different rollouts using the same notion of runtime. It is not obvious that parallel updating using partially propagated states will provide benefits.