# OpenReview forum: "Improving Anytime Prediction with Parallel Cascaded Networks and a Temporal-Difference Loss"
_NeurIPS.cc/2021/Conference — NeurIPS 2021 Poster_

### Official Review · Reviewer_Qwk8 · 2021-07-05

**Rating:** 8
**Confidence:** 3

**Summary:**

The authors propose to tighten the link with biological neural networks by designing a cascaded network that is more compatible with the dynamics of information processing observed in the brain. In cascaded networks, all layers process the information in parallel and are fed with an updated input at each time step. The authors also introduce a loss, the Temporal Difference, that improve the performance of cascaded networks. The authors demonstrate that the cascaded network offers better trade-off accuracy and better noise robustness than serial one. In addition, the authors demonstrate that class instance that are rapidly recognized tend to be more prototypical in cascaded network

**Limitations And Societal Impact:**

The authors did not provide any insight on 'Limitations and Negative Impact'

**Main Review:**

## General comments :

The paper offers an interesting and novel perspective on the dynamics of neural network. The various experiments are all very relevant, and illustrate well the message of the paper. The two concerns I have are relatively minors : i) I found the paper hard to read because many sentences have just no meaning. Now that the rush of the submission is behind, the authors should take time to carefully re-read the article and correct the typos and also be sure that all the sentences has an English meaning. ii) It would have been interesting to confirm the results of the paper on a higher resolution database (e.g. imageNet).

Despite these 2 comments, I think the presented article is novel enough and should be useful for both the Neuroscientist and Machine Learning Community. This is in general a good article ( My score : 8).

## More specifics comments :
* Introduction : The concept of anytime prediction should be defined before the section ‘related work’ as it is crucial to understand the origin of the idea behind the paper. I would also suggest to define it before the introduction of figure 1b, which is a good example of an algorithm doing anytime prediction

* Figure 1. The figure 1 is very important to well understand the concept of the paper but is not clear enough on my mind. The color are not well chosen and a color legend is necessary to well grasp the message of the paper.

* Section : ' Deep Cascaded Network'. You said the batch norm has been changed to include ’time-step-conditional normalization’, but there is no further explanation. Could you please explain what is a time-step-conditional normalization ?

## Typos
Here are some of the typos I found (I quickly give up the idea to note all the typos because of their numbers)

Line 148 : ’The choice of alpha over a had no impact on our findings …’ —> ’ The choice of alpha had no impact on our findings

Line 185 : ‘lambda = 0 corresponds with requiring a prediction only of the wether on the next day’. This sentence has to be corrected

Line 219 : ’SINGLEHEADfor’ —> ’SINGLEHEAD for'

Line 304 : ‘an then the noise-free is presented input until … ‘. This sentence has to be corrected


**Time Spent Reviewing:**

4-5 h

---

> ### Author Response · Authors · 2021-08-10
> **Response to reviewer Qwk8**
>
> We thank the reviewer for their thoughtful and helpful review, and we apologize for any difficulties resulting from our writing style.
>
> **Concerns with writing and style.** Two authors, both native English speakers, have carefully read through the article and a spelling / grammar checker has been used, and we can find only one additional typo (a missing “the” in the Figure 1 caption) beyond the 3 that the reviewer identified. (The 4th issue that the reviewer flagged may be a confusion about the word “weather”, which refers to conditions outdoors such as temperature, not the conjunction “whether”. While the sentence is grammatical, it is awkwardly phrased and we have rewritten it.) If the reviewer’s frustration with the text is due to sentence constructions, we would appreciate the reviewer's guidance to identify specific problematic phrases, sentences, or paragraphs. We strive to make our work clear and accessible to all readers.
>
> **Higher resolution database.** Following your suggestion and that of another reviewer, we ran an additional simulation with a subset of ImageNet classes. We trained a model on 10 breeds of terriers (dogs) using 224x224 ImageNet data. These breeds are visually similar to one another and cannot be discriminated perfectly based on any simple feature such as color. We examined speed-accuracy curves (Fig 4 of our manuscript) and the prototype effect (Fig 2). We observed the same qualitative behavior from this new model as we observed for our models trained on smaller resolution images. To provide one concrete result, we obtain an asymptotic test accuracy of 67.2% for CascadedTD(0) versus 63.8% for SerialTD(1)-MultiHead (i.e., Shallow-Deep Nets), and a 13% speed up for CascadedTD(0) to reach a threshold of 50% accuracy. We will include this experiment in our final manuscript and also conduct a sweep of TD $\lambda$ values (Fig 3). [For training details, each class had 1300 examples in ImageNet, which were split 90/10 into train and test examples. All simulation details were the same as our previous work, except batch size had to be reduced to 32 from 128 for the larger images, and as is standard for ImageNet, the first convolutional layer had 7x7 receptive fields.]
>
> **Defining anytime prediction.** We should indeed have defined the term early in the paper. We have inserted a definition following the first use of the term (line 51). The definition roughly is: "In a model where computation is broken up into a sequence of primitive sequential operations and completion of the computation may require $n$ operations, an anytime prediction model will produce a read-out after each of the operations. In a sequential model (like ordinary neural nets), each operation corresponds to propagating activation completely through one layer or one block; in a cascaded model, each operation corresponds to propagating (delayed or partial) activation through all layers or all blocks”.
>
> **Concerns about Figure 1.** We have re-done Figure 1 with arrows coming from the output heads (the trapezoids), along with the label ‘readout’. We have also used larger font sizes to allow the reader to see the time indices (t=1, t=2, etc.). And we have rewritten the caption for subfigures (b) and (c), and we would appreciate the reviewer’s feedback if this rewrite is clearer:
> > (b) A standard  _serial_ ResNet is unrolled in time, with columns depicting time slices. Each rectangle is a ResNet block, which may consist of two or more convolutional layers. In the serial model, blocks are updated sequentially.  Blocks which have not yet been activated are colored white and blocks which have been activated are shown in a hue unique to that block. The input is cyan. The narrow bars within each block signify the activation state of all blocks below that are contributing to the block's state (via skip connections). Read out from the model is via the yellow trapezoid at the top, which enables anytime prediction. The narrow bars inside the trapezoid indicate the block information available at each time for classification (via skip connections). (c) A _cascaded_ ResNet is unrolled in time. In the cascaded model, all blocks update in parallel; however, at each step, they may rely on partial updates of lower blocks. As a result, multiple processing steps are required for a layer's activation to reach its asymptotic state. The color intensity (saturation) of a block indicates how close a block's activation state is to its asymptotic state.
>
> **Time-step-conditional batch norm.** Although a further explanation of this term is presented in the supplementary materials, we failed to point the reader to this explanation. We have corrected the paper. To explain: Because we have a model that is unrolled in time, a given batch norm operation will be replicated for each time step. In principle, either we could perform a separate batch norm operation at each time step using the activations at that time step or we could perform a single batch norm operation combining the activations across steps. We do the former.

---

### Official Review · Reviewer_gw8x · 2021-07-12

**Rating:** 6
**Confidence:** 5

**Summary:**

This paper proposes to use skip connections in feedforward artificial neural networks to integrate information over multiple time steps by defining a delay for each connection in the network. Extensive experiments with a ResNet18 architecture show that this execution plan leads to an increase in early prediction accuracy when using anytime predictors, to faster predictions when samples are closer to a class prototype, more robustness to noise and a better out-of-distribution (OOD) detection compared to a sequential architecture. Temporal difference learning (TD) is utilized to train the neural network and an analysis is done on CIFAR-10, CIFAR-100 and TinyImageNet to show that the often used value of lambda=1 is suboptimal for these kinds of tasks.


**Limitations And Societal Impact:**

Limitations and societal impact were not discussed in detail.


**Main Review:**

Overall, the theory is well explained and the experimental work meticulously shows how the combination of streaming rollout, temporal difference learning and anytime prediction leads to an improved network compared to sequential execution. While being very similar idea-wise to Fischer et al ([10] in the paper), additional experiments extend the previous work and undermine all claims made in the introduction.

Still, a few things are unclear after reading the paper.
1. I think the work of Fischer et al would have to be discussed in a bit more detail, as the method is very similar in the way that OSD in this paper is exactly the streaming rollout in Fischer et al. There is also follow-up work on streaming rollouts with DenseNets, presented in Kugele et al [1], that is not discussed in the paper.

2.  When scrutinizing TD(lambda) in your experiments, it is mentioned that TD(1) risks to overfit your test set. It is not clear to me, what this means (I guess you mean overfitting on the training set, leading to a worse performance on the test set?), and more importantly, why this should be the case.

3. When introducing Deep Cascaded Networks, it is not clear how exactly the readout for the anytime predictions is done. Are there 9 blocks and 9 time steps, and at each time step t, block output t is read out to make a prediction?

4. In Fig. 4: Why should multi-head be worse? In the worst case, can still learn that all heads are the same. Maybe another explanation could be overfitting?

5. In Tab. 1: As this is in my opinion a non-trivial measure, it would have been good to explain the coefficient in even more detail and how it relates to the results.

6. Generally, it would have been interesting to see experiments with other architectures, e.g. the mentioned U-Net, DenseNet or Transformers.

7. A comparison between EWS and OSD in terms of computational complexity (especially in the backward pass) would have been interesting as well. If I am not mistaken, gradient descent would lead to the problem that it has to backpropagate through all time steps in the case of EWS.

In summary, this paper is a relevant contribution to improve the understanding of rollouts. I think this paper is on the borderline of being ready for publication and I hope some of my concerns are addressed in the rebuttal.

Small comments that do not influence my score:
- ln 149: seems like there are words missing
- Informal language: ln 213: Let's step; ln 78: home in -> converge
- ln 304: the word "input" has to be before "is presented"
- Overall it looks like the official style file is not used (e.g. sections are not numbered)


[1] Efficient Processing of Spatio-Temporal Data Streams With Spiking Neural Networks, A. Kugele, T. Pfeil, M. Pfeiffer, E. Chicca, Front. in Neuroscience, Vol 14 (2020)


------
After reading the other reviews and rebuttals, I increased my score from 5 to 6.

**Time Spent Reviewing:**

4.5

---

> ### Author Response · Authors · 2021-08-10
> **Response to reviewer gw8x**
>
> We thank the reviewer for their objective and constructive review.
>
> **Relationship of our work to Fischer et al. and Kugele et al.** We should indeed have given more discussion to Fischer et al. and we were unaware of Kugele et al.'s relevant and interesting work. Fischer et al. present a general taxonomy that includes our feedforward cascaded model, but their focus is almost entirely on _formal definitions and a framework_ that lays out the space of all well-formed roll out patterns (update orders). They present some basic simulations using pretrained models that are rolled out in different orders. In contrast, our focus is almost entirely on _training procedures_ to leverage the dynamics of cascaded models, on early read-out mechanisms, and on the computational consequences of these training and read-out mechanisms. Kugele et al. are focused on spiking neural net dynamics, on time-varying inputs (as does Carreira et al. [our ref 5]), and on reductions in latency that are obtained as a sequence unfolds due to autocorrelations in the input sequence. (Coincidentally, we ran a series of similar experiments with video sequences, but decided to omit them from our submission for lack of space.) Notably, Kugele et al. explore a variety of heuristic training losses and they settle on TD(1) as their preferred loss, but they do not explore the rest of the TD($\lambda$) family nor how training influences the nature of representation in the model (see Sec 3.2 of their paper; their $a_k = 1$ corresponds to TD(1)).  In summary, our work is complementary to these related papers; each makes its own distinctive contribution. We will update our related literature section to include these remarks.
>
> **Concerning the choice of $\lambda$ used in experiments.** Our wording on lines 188-189 is confusing. We wrote that choosing $\lambda=0$ avoids the “risk of overfitting to our test set.” We were trying to say that picking the lambda that yields the peak of the Figure 3 curves would be overfitting to our test set. For simplicity, we chose to use $\lambda=0$ everywhere, even though model performance might potentially be improved by treating $\lambda$ as a hyperparameter.
>
> **Read out for anytime predictions.** Figure 1 and lines 193-194 describe the read out procedure. Since our submission, we have improved Figure 1 to indicate that the output heads (trapezoids in Figure 1) generate an output which we have labeled as ‘readout’. In addition, we have rewritten the caption to Figure 1, which is copied in the response to reviewer Qwk8. The read-out procedure is not what you are hypothesizing: read out at each step _t_ is from the output head associated with step _t_, which incorporates input from all active blocks (Figures 1b,c).
>
> **Why MultiHead yields lower performance.**  We agree with the reviewer’s assessment that overfitting is to blame. There is nine times as much data for training the single head as there is for training each of the 9 step-specific heads. Although the 9 separate heads in principle have greater flexibility, that flexibility is probably not needed because the additivity of the residual activations in the ResNet architecture encourages all blocks to express their results in the same representational space.
>
> **Table 1.** Thank you for the suggestion to better explain the correlation measure. Lines 262-267 attempt to clarify the role of the correlation coefficient, but we will do so even more explicitly in the table caption.
>
> **Comparing EWS and OSD.** The space and time complexity of training either with EWS or OSD is linear in the number of steps that the model runs for. Because EWS stretches the number of steps, it would indeed be costly to train with EWS, as the reviewer notes. However, we always train with OSD for efficiency’s sake and then evaluate the model with EWS. We’ve found that this procedure leads to very similar behavior from EWS and OSD, except the time course of EWS is smoother which allows us to measure small differences in model run time.

---

### Official Review · Reviewer_i5tq · 2021-07-16

**Rating:** 7
**Confidence:** 4

**Summary:**

Feedforward networks can be converted into networks with temporal cascaded dynamics by introducing a propagation delay and re-training the model with a new loss. Via the use of skip connections, such cascade models can trade-off inference time and make predictions on prototypical images more quickly.


**Limitations And Societal Impact:**

Yes

**Main Review:**

**Strengths**:
* The proposed cascade model improves over a previous model in terms of its time/accuracy tradeoff, might be more robust to dealing with certain kinds of noise, and intuitively takes longer to recognize difficult images
* The authors analyze the proposed model and posit that it classifies prototpical images more quickly than images in unusual views
* The paper takes a step to implementing meta-cognition with an additional model inspecting the time-dynamics of the cascade model (although this part comes quite short)

**Criticisms**:
* The proposed cascade model in its current form is limited in its generality: Analyses have only been run on datasets with small image inputs (CIFAR, TinyImagenet) and might not generalize to more natural images such as ImageNet
* Potential benefits of quick inference for prototypical images are never shown directly with respect to runtimes: one of the primary implied benefits of anytime prediction is the availability of initial class predictions after only little time has passed. This should be shown quantitatively, e.g. as a function of CPU cycles or on (simulated) neuromorphic hardware (e.g. Intel Loihi). Otherwise the paper is left with demonstrating that the proposed CascadedTD model improves over time, but it is unclear how its fast/slow predictions compare with a standard feedforward architecture.

Minor:
* the paper uses the brain as motivation, but never compares against any brain data. I would personally like this to be made more clear upfront.
* scholarship: the way the text is written still implies this to be the first model with time delays between neural transmissions. E.g. https://arxiv.org/abs/1804.10123 and https://www.biorxiv.org/content/10.1101/408385v1 address this very idea and are not mentioned in the paper (I pointed this out in an earlier review, so it's quite frustrating to still not see these papers mentioned/discussed. However I did notice improvements in the connection to neuroscience results which I am happy to see.)
* declaring using lambda=0 in line 187 and then choosing lambda=1 for SerialTD and SerialTD-Multihead is a bit confusing, I think one of these needs to be changed. I realize line 187 refers to only CascadedTD, but that choice was still surprising to me, especially given the improved performance of SerialTD(0) in Fig. A3

EDIT after rebuttal: The authors have responded with good quantitative arguments and laid out conceptual ideas more clearly. These revisions will make for a good paper, I have updated my score 5 -> 7.

**Time Spent Reviewing:**

2

---

> ### Author Response · Authors · 2021-08-10
> **Responses to reviewer i5tq**
>
> We thank the reviewer for the constructive feedback. Responses to specific comments follow.
>
> **Concern that results might not generalize to high resolution images.** Although we are unaware of phenomena in the deep-learning literature that are observed with high resolution images (ImageNet: 224x224) but not with intermediate resolution images (TinyImageNet: 64x64), we found the reviewer’s suggestion valuable and we ran an additional simulation with a subset of ImageNet classes. We trained a model on 10 breeds of terriers (dogs) using 224x224 ImageNet data. These breeds are visually similar to one another and cannot be discriminated perfectly based on any simple feature such as color. We examined speed-accuracy curves (Fig 4 of our manuscript) and the prototype effect (Fig 2). We observed the same qualitative behavior from models as we observed for our models trained on smaller resolution images. To provide one concrete result, we obtain an asymptotic test accuracy of 67.2% for CascadedTD(0) versus 63.8% for SerialTD(1)-MultiHead (i.e., Shallow-Deep Nets), and a 13% speed up for CascadedTD(0) to reach a threshold of 50% accuracy. We will include this experiment in our final manuscript and also conduct a sweep of TD $\lambda$ values (Fig 3). [For training details, each class had 1300 examples in ImageNet, which were split 90/10 into train and test examples. All simulation details were the same as our previous work, except batch size had to be reduced to 32 from 128 for the larger images, and, as is standard for ImageNet, the first conv layer had 7x7 receptive fields.]
>
> **Comparisons of run times of Cascaded and Serial models.** Let us try to convince the reviewer that key results in our paper (e.g., Figure 4) in fact provide a quantitative comparison on simulated parallel hardware. We didn’t do a great job expressing this point in the current draft of the paper. If the reviewer considers our response sensible, we will include it in a revision of the paper.
>
> The premise of our work is that we have massively parallel hardware with delays on inter-component communication. This architecture allows all ResNet blocks to be updated simultaneously.  A block update involves a sequence of matrix multiplications, vector additions, and vector thresholding operations. Because all model variants perform block updates, we needn’t break down the block update into its primitive operations; instead, we consider the basic cycle to be a block update. Due to the assumed communication delay in our hardware, the updated output from a block is not available to other blocks until the next cycle. Both the Serial and Cascaded models can be run on this parallel hardware. The Serial model does not take full advantage of the parallel hardware because it updates only block _k_ at cycle _k_. The Serial model performs anytime  read out at cycle _k_ by summing the outputs of blocks 1 through _k-1_ and passing the sum through the classifier head. The Cascaded model takes full advantage of the parallel hardware by updating all blocks at each cycle _k_ using their output from cycle _k-1_. Just as the Serial model, the Cascaded model can perform any time read out using the one-cycle lagged block outputs. Thus, comparing the Serial and Cascaded models in units of block updates--as we have done throughout the article--does provide a runtime comparison on the assumed parallel hardware. (Note that Fischer et al. [our ref 11] similarly compare models with different rollouts using the same notion of runtime.) The reviewer’s suggestion to compare based on existing neuromorphic hardware is intriguing, but any such comparison would not address the key question we ask, which is whether consideration of the novel hardware architecture we propose is warranted. It is far from obvious that parallel updating using partially propagated states will provide benefits. And the temporal-difference procedure we explore is valuable not only on the parallel hardware, but as we have shown, for existing anytime prediction models.
>
> **Request that we be more clear upfront that the brain is used as motivation but modeling brain data is not the goal.** Lines 41-43 of our submission read: “Fundamentally, our investigation asks: Supposing we take a step toward biological realism with massively parallel hardware and relatively slow inter-neuron communication, what are the computational benefits and consequences?” Because the reviewer found this inadequate, we will add an additional footnote stating: “Like much other research in deep learning [citations], biology informs our work by providing novel forms of inductive bias. Our goal is to investigate computational consequences of these biases, not to model biological phenomena per se.”
>
> **Text implies Cascaded model is the first to consider time delays in neural models.**  Lines 87-97 of the manuscript describe work, from the 1980s to the recent deep-learning literature, on cascaded neural architectures. We would appreciate the reviewer’s guidance to point us to where our text implies we are the first.
>
> **Missing references.** In response to the reviewer’s review of our work from a previous conference (thanks for taking on this paper again!), we added a paragraph (lines 98-101) distinguishing our approach from serial RNN models for vision. We cite five such papers, but failed to include the two specific citations mentioned by the reviewer. We will add these citations. We do cite other papers by the CORnet authors. IamNN, which is a ResNet with weight constraints across layers, is operationally an unfolded-in-time serial RNN. IamNN is similar to our refs 34 and 52, which are RNNs used for anytime prediction.
>
> **Choice of $\lambda$ for various models.** As we explain in the footnote on p. 6, we chose $\lambda=1$ for SerialTD and SerialTD-Multihead in the main paper because these two models were meant to correspond to current practice, which involves the combination of serial block updating and TD(1) training. (Shallow-Deep Nets are equivalent to SerialTD-Multihead with $\lambda=1$.)  In the footnote, we point the reader to the appendix, where Figure A.3 shows that using $\lambda=0$ does strictly improve the speed-accuracy trade off for Serial models. However, we note that even with $\lambda=0$, the Serial models have asymptotically lower performance and an inferior speed-accuracy trade off relative to CascadedTD(0). We thus did not feel we introduced confounds by relegating the exploration of $\lambda=0$ for Serial models to the appendix (due to space considerations). We thank the reviewer for their careful attention to the appendix, and we welcome suggestions for how we might change the text to address the confusion the reviewer experienced; we have been as explicit as possible about the issue without complicating Figure 4 further by showing twice the number of curves (for all combinations of $\lambda=0$ and $\lambda=1$).

---

> > ### Comment · Reviewer_i5tq · 2021-08-25
> > **concerns addressed**
> >
> > Thank you very much for the extensive response and investing additional efforts to test this model on a subset of the ImageNet benchmark.
> >
> > **Generalization to high resolution images**: The added results are convincing to me, well done.
> >
> > **Comparisons of run times of Cascaded and Serial models**: Thank you for laying out the intuition and the connection to hardware runtimes more clearly, I had indeed not appreciated the block updates as a proxy measure. Making this more clear in the revised text and figure labels/captions should help readers make quicker sense of this than I did.
> >
> > **Other points**: For all other points, I am satisfied with the planned changes / reasoning behind choices.
> >
> > The authors have responded with good quantitative arguments and laid out conceptual ideas more clearly. These revisions will make for a good paper, I have updated my score 5 -> 7.

---

> > > ### Author Response · Authors · 2021-08-25
> > > **Thank you**
> > >
> > > Thank you for taking the time to provide us with helpful feedback and for considering our response. We will certainly update the paper to clarify the runtime issue as well as other points you raised.

---

### Decision · Program_Chairs · 2021-09-27

**Decision:**

Accept (Poster)

**Comment:**

This paper presents a novel ANN architecture (which the authors term "cascaded" networks) inspired by the real brain. In brief, this architecture introduces delays in the propagation between computational blocks, such that skip connections serve to provide a rapid inference, whereas over time, the effective depth of the architecture increases. They use this architecture for anytime prediction, and develop a temporal difference based training algorithm that encourages the networks to find rapid solutions to the inference problem when possible. They show that this approach provides some interesting and desirable properties, such as different "reaction times" to prototypical versus atypical samples and greater robustness.

The initial review scores were a mix. Nonetheless, the reviewers were generally positive, although there was some concerns about the generality and potential benefits of the approach for neuromorphic systems. However, after author responses and discussion, though there was still some divergence in scores, the reviewers largely agreed that this paper makes a sufficiently interesting and worthwhile contribution to the field for acceptance.